# The Adenovirus Dodecahedron: Beyond the Platonic Story

**DOI:** 10.3390/v12070718

**Published:** 2020-07-02

**Authors:** Solène Besson, Charles Vragniau, Emilie Vassal-Stermann, Marie Claire Dagher, Pascal Fender

**Affiliations:** 1Centre National de la Recherche Scientifique, Université Grenoble Alpes, Commissariat Enérgies Alternatives, Institut de Biologie Structurale, 41 rue des Martyrs, 38042 Grenoble, France; solene.besson@ibs.fr (S.B.); charles.vragniau.phd@gmail.com (C.V.); emilie.stermann@ibs.fr (E.V.-S.); marie-claire.dagher@ibs.fr (M.C.D.); 2 Centre National de la Recherche Scientifique, Université Grenoble Alpes, Institut National Polytechnique Grenoble, Technique de l’ingénierie Médicale et de la Complexité, TIMC-IMAG Bât Jean Roget Faculté de Médecine et Pharmacie, 38700 La Tronche, France

**Keywords:** adenovirus, dodecahedron, platonic solids, virus like particles, receptors, vaccines, structure, viral spreading

## Abstract

Many geometric forms are found in nature, some of them adhering to mathematical laws or amazing aesthetic rules. One of the best-known examples in microbiology is the icosahedral shape of certain viruses with 20 triangular facets and 12 edges. What is less known, however, is that a complementary object displaying 12 faces and 20 edges called a ‘dodecahedron’ can be produced in huge amounts during certain adenovirus replication cycles. The decahedron was first described more than 50 years ago in the human adenovirus (HAdV3) viral cycle. Later on, the expression of this recombinant scaffold, combined with improvements in cryo-electron microscopy, made it possible to decipher the structural determinants underlying their architecture. Recently, this particle, which mimics viral entry, was used to fish the long elusive adenovirus receptor, desmoglein-2, which serves as a cellular docking for some adenovirus serotypes. This breakthrough enabled the understanding of the physiological role played by the dodecahedral particles, showing that icosahedral and dodecahedral particles live more than a simple platonic story. All these points are developed in this review, and the potential use of the dodecahedron in therapeutic development is discussed.

## 1. Introduction to Adenoviruses

Adenoviruses are non-enveloped viruses with a diameter between 70 and 90 nm, presenting an icosahedral capsid. They possess linear double stranded DNA with inverse repeated sequences at each end and an encapsidation sequence. The *Adenoviridae* family is divided into 5 genera: Mastadenoviruses, Aviadenoviruses, Atadenoviruses, Siadenoviruses, and Ichtadenoviruses. They can infect a large variety of species, such as cows, dogs, ducks, horses, snakes, fish, and humans (https://sites.google.com/site/adenoseq/). The human adenoviruses (HAdVs) are part of the Mastadenovirus group. These human serotypes have been sorted into seven species from A to G depending on their biological, genetic, biochemical, and structural properties (http://hadvwg.gmu.edu). Human adenoviruses are responsible for fever, infections of respiratory tracks, gastroenteritis, and conjunctivitis. In most cases, the symptoms are not visible in healthy persons [1]. However, some serotypes can cause more severe infections leading in some cases to death in immune-deficient patients and infants. For example, HAdV3, HAdV5, and HAdV7 are responsible for acute respiratory tract infections, and a recent outbreak of HAdV7 in a New Jersey rehabilitation center resulted in eleven deaths (https://www.nj.gov/health/cd/topics/adenovirus.shtml). HAdV40 and HAdV41, on the other hand, are known to cause acute and persistent gastroenteritis in children [2]. Subgroup D adenovirus serotypes are known for their ocular tropism resulting in conjunctivitis or epidemic Keratoconjunctivitis (EKC) [3].

All adenoviruses are composed of three main capsid proteins or ‘capsomers’: the hexon, the penton base, and the fiber (Figure 1). The main protein is the hexon, with 240 trimeric capsomers forming the 12 triangular facets of the icosahedral viral capsid [4,5]. The hexon protein is therefore the most abundant structural protein of the adenovirus since 720 hexon copies of the monomer (~110 kDa) are displayed on the virus surface, and it represents 80% of the virus total mass [6]. The penton base is an homopentamer composed of five ~60 kDa monomers displayed on each of the 12 viral capsid apexes. Therefore, each monomer is presented 60 times on the viral capsid. The penton base contains 2 hypervariable loops exposed at the virus surface and named variable loop and RGD (Arginin-Glycin-Aspartic Acid) loop [7]. The highly conserved RGD motif has been described to interact with the αvβ3 and αvβ5 integrins, inducing virus internalization [8,9,10]. However, some adenoviruses, such as HAdV40 and HAdV41, lack this motif, although this does not prevent them from infecting cells [11]. Recently, it has been shown that those serotypes nevertheless interact with 6-containing integrins with a similar affinity [12].

The fiber, like the penton base, is present on each of the 12 viral capsid apexes. It is an homotrimer composed of a tail, a shaft and a knob domain (Figure 1). The tail, corresponding to the N-terminal of the fiber, forms a non-covalent interaction with the penton base. The shaft contains repeated patterns of 15 amino acids [13,14]. The number of repetitions gives the shaft a length which varies from one serotype to another. For example, the shaft measures 9 nm for HAdV3 and 36 nm for l’HAdV40. The knob corresponds to the globular C-terminal of the fiber and interacts with receptors, allowing virus attachment to the cells [15]. Therefore, the fiber’s knob dictates the viral tropism. To date, three main protein receptors have been identified: CAR, CD46, and desmoglein (DSG)2 [16,17,18]. In addition to CAR, most of the subgroup D adenoviruses can also use non-protein receptors, such as sialic acids or the GD1-glycan [19,20].

The non-covalent complex formed by the penton base and the fiber is called the penton. This complex contains all the information needed for virus attachment and internalization. Its formation is due to the interaction between a conserved motif (FNPVYPY) present in the fiber’s tail, and a complementary sequence located at the interface between two neighboring penton base monomers [7]. This penton is the building block of the adenovirus dodecahedron described below.

## 2. The Platonic Solids

A platonic solid is a regular convex polyhedron meaning that all angles are the same and all the sides are equal in length. It is made of congruent (identical in shape and size), regular polygonal faces, such as equilateral triangles, squares, or pentagons, meeting at each vertex (Table 1). Furthermore, these solids can be inscribed into a sphere. Five solids meet these criteria: the tetrahedron (4 triangular faces), the cube (6 square faces), the octahedron (8 triangular faces), the dodecahedron (12 pentagonal faces), and the icosahedron (20 triangular faces). These solids are named after the Greek philosopher Plato, who theorized that the classical elements of the world were made of these 5 solids. Plato associated each solid with each of the 4 basic elements (earth, air, fire, water) and reserved the fifth for the universe or heaven (https://www.britannica.com/science/Platonic-solid). They can be associated in dual pairs, where the solids in a dual pair have the same number of edges, and the number of vertexes of one corresponds to the number of faces of another. Therefore, the tetrahedron is self-dual, while the cube and the octahedron form a dual pair, and the dodecahedron and icosahedron form another (Table 1).

The two solids forming a dual pair can be inserted into one another. These solids manifest in the world around us, for example in crystals and in viruses. Indeed, some viruses, like the adenoviruses, possess an icosahedral viral capsid. Moreover, during the adenoviral natural replication cycle, dodecahedrons can be produced depending on the adenovirus serotype. Dodecahedrons and icosahedrons are dual, therefore the adenovirus dodecahedron being an icosahedron when focusing on the vertices and a dodecahedron when focusing on the faces (Table 1).

## 3. Discovery of the Adenovirus Dodecahedron

### 3.1. Discovery and Spontaneous Production

The adenovirus dodecahedrons were first discovered in Sweden in 1964. While purifying soluble antigens produced during an HAdV3 infection of human cells, Erling Norrby observed a homogenous population of particles sedimenting by centrifugation at a rate corresponding to 50–60 S. They appeared as six- or five-pointed ‘stars’ composed of identical tubular, capsomer-like structures, each associated with a thin club-shaped projection [21]. The overall diameter between the points surrounding the star was assessed between 40 and 50 nm. These points were later attributed to the fiber’s knobs, and the tubular capsomer was ascribed to the penton base forming the core (star) of the dodecahedron. These particles do not contain hexons, but are exclusively made of penton base and fiber proteins.

Other studies have shown that not all HAdV serotypes are able to produce dodecahedrons. This has been shown to be the case during the replication cycle of serotypes 4, 7, 9, 11, and 15 [22,23,24,25]. In contrast, there is no evidence that serotypes 1, 2, 5, 6, 12, 16, 40, and 41 can produce such dodecahedrons [26,27,28,29]. It is also worth noticing that none of the well-studied subgroup C serotypes are able to produce dodecahedrons and that, in subgroup B, dodecahedron production is serotype-dependent (Table 2).

The most studied adenovirus dodecahedron (Ad-Dd) is derived from HAdV3. Like all adenoviral capsid proteins, the penton base is synthesized in the cytoplasm (Figure 2a). This was shown by immunofluorescent of the penton base (i.e., the dodecahedron): the signal can be detected about 12 h after the virus internalization. The penton base protein is then transported to the nucleus, where it accumulates and likely dodecamerizes [30]. At 20 h post infection, immunofluorescence indicates a substantial concentration around the internal nuclear membrane, whereas the progeny virions crystalize randomly in the nucleus as observed by electron microscopy on ultrathin sections. No function has been attributed yet to dodecahedrons in the nucleus, but it would be interesting to investigate whether this specific internal perinuclear location could be associated with a physiological process, such as viral mRNA export. Nevertheless, the number of dodecahedral particles produced per human cell infected by HAdV3 in culture was estimated to be about 7.5 × 10^5^, which is much more than the number of infectious virions, reinforcing the idea that this particle does play a role in the viral cycle. This point is further discussed later in this review (see Section 6).

### 3.2. Recombinant Dodecahedrons

The adenovirus dodecahedron was ‘rediscovered’ in 1997 by co-expressing the penton base and the fiber of HAdV3 in insect cells using the baculovirus expression system. Insect cells lysates were then subjected to a sucrose gradient and the heavy fractions were analyzed by negative stain electron microscopy, highlighting spherical particles arranged into regular dodecahedrons by their bases [31]. These particles display two-fold, three-fold, and five-fold symmetry axes. It is worth noticing that, as expected, the HAdV-2 penton base expressed in a similar way did not form dodecahedrons but only pentameric penton bases, as HAdV-2 does not form dodecahedrons under ‘natural’ conditions. This confirms that not all adenovirus serotypes are able to generate such particles. The recombinant dodecahedron made of 12 pentons (12 penton bases with 12 protruding trimeric fibers) was called the Penton Dodecahedron (Pt-Dd) (Figure 2b). Since only those two genes were cloned in the baculovirus vector, it proved that no other adenoviral component was needed for the formation of this particle. Moreover, the authors showed that the expression of the penton base alone resulted in the production of the dodecameric core, called the Base Dodecahedron (Bs-Dd). This observation showed that the penton base itself contains all the structural information required for dodecamerization and that the fiber was not required. An important feature of these two particles (Pt-Dd and Bs-Dd) is their high capacity to enter cells (Figure 2c). It was also shown that, if the internalization of the Pt-Dd containing the two proteins involved in adenovirus entry was reported as expected, then the fiber-devoid Bs-Dd could also efficiently enter into cells, but a 10-times higher concentration was needed to achieve a similar level [32].

Pseudotyping involves adding a viral component onto another viral or pseudoviral scaffold. Since the penton base/fiber interaction mechanism is conserved between several adenovirus serotypes [7], fiber-pseudotyped dodecahedrons are produced. This has been demonstrated by the in vitro addition of the long HAdV2 fiber on Bs-Dd [31] or by coexpression of the HAdV3 Bs-Dd with the ‘short’ fiber of the enteric serotype HAdV41 in insect cells [29]. This result elegantly showed that fibers from serotypes, for which the penton base cannot dodecamerize (HAdV2, HAdV41), can be artificially dodecamerized on the HAdV3-derived Bs-Dd scaffold (Figure 3).

## 4. Structural Determinants of Dodecamerization

When they form a dodecahedral morphology, the penton bases interact with each other, whereas, when they are on the viral facets, they are separated by the hexons. To the best of our knowledge, this is an exception in virus-like particles (VLPs) [33], which are generally made up of proteins that interact in the same way as they do in the viral capsid. Moreover, as mentioned above, not all serotypes are capable of producing such particles, suggesting that there must be a particular molecular pattern that is responsible for the dodecamerization phenomenon. Several structural studies on dodecahedrons have been undertaken in order to understand the molecular basis for this. Preliminary work was carried out using cryo-microscopy in 1996 using dodecahedrons with and without the fiber. Despite the low resolution accessible by cryo-EM at that time (20–25 Å), the existence of an internal cavity of ~350 nm^3^ in the center of the particle was revealed [34]. In addition, it was reported that the fixation of the fiber caused a minor density change at the surface of the penton bases.

In 2005, the structure of the Ad2 base was reported by X-ray crystallography at 3.3 Å [7]. Remarkably, this protein, which is unable to dodecamerize under physiological conditions, crystallized in a dodecahedral arrangement in the presence of 1.6 M ammonium sulfate and 10% dioxane. By combining this discovery with an improvement in the dodecahedral resolution of HdAdV3 dodecahedrons by cryo-EM (9 Å), a quasi-atomic structure of the Ad3 dodecahedral particle was reconstructed [35]. In this work, three critical regions involved in dodecamerization were identified in the primary penton base sequence. The first one is located close to the N-terminal (58-SELS-61) and allows homotypic interaction in the two-fold axis with the same sequence from another monomer from an adjacent pentamer. The two other regions (98-NNDFT-102 and 424-FRSTSQ-429, Figure 4a) are distant from each other in the monomer primary sequence but form heterophilic interactions with their counterpart in the monomer of an adjacent pentamer (Figure 4b). Later on, the first crystallographic structure of the Ad3 dodecahedron at 3.8 Å confirmed these data and showed that the charged residues D100 and R425 (highlighted in bold above) formed a salt bridge, as well as an additional network involving hydrogen bonds with N98 and T427 (underlined above), thus stabilizing the structure [36]. Unexpectedly, those residues were not exclusive to HAdV3 but were also present in the non-dodecahedron forming (HAdV2) penton bases. This suggested that, even though those regions are involved in dodecahedron formation, another critical mechanism is required for particle stabilization. The authors elegantly showed that an N-terminal strand-swapping between neighboring HAdV3 penton bases occurred, thus locking the particle. HAdV2 penton bases, on the other hand, were not capable of this. This N-terminal interlocking is controlled by region A (58-SELS-61) since its mutation results in a drastic decrease of dodecahedron stability [36].

## 5. A Tool for Receptor Identification and Characterization

The particle size and multivalency of dodecahedrons make them extremely interesting tools for identifying and characterizing receptors or cellular partners interacting specifically with the penton base or the fiber. The first example was provided with heparan sulphate proteoglycans (HSPGs). Using cells which either expressed or did not express HSPGs, it was shown that both Pt-Dd and Bs-Dd could interact specifically with these cell surface glycoproteins [37]. This result was unexpected since it had previously been reported that, unlike other adenoviral serotypes, HAdV3 could not use these molecules in the same way [38]. This observation suggested either that the penton bases create electrostatic surfaces by stacking in the dodecamers, therefore allowing the interaction with HSPGs, or that the presence of hexons in the viral capsid has a negative effect on this interaction. In any case, it is interesting to note that these particles, partly mimicking the virus, acquire a different property from the virus from which they are derived. The fiber does not seem to be involved in the interaction with HSPGs, since both Bs-Dd and Pt-Dd interact with an apparent affinity on the nanomolar scale. These data obtained by surface plasmon resonance (SPR) on immobilized heparin probably reflect the high avidity due to the multivalency of these particles. A few years later, it was shown by cryo-EM that an extra density was indeed visible on the RGD loop of dodecahedron incubated with an HS oligosaccharide (dp8 = 4 sulfated di-saccharides). A basic BxBB pattern (where B stands for basic, and x for any other amino acids) close to the RGD sequence could play a role in the regulation of the interaction with integrins [39]. The mechanism by which HSPGs regulate this interaction was not fully addressed, but a ‘click to fit’ mechanism was proposed, meaning that a structural change triggered by HSPG could modulate the affinity of the penton base for integrins.

A second example is the identification of several intracellular partners interacting with these dodecahedral particles. By screening a gt11 phage library encoding cDNAs from human lung cells with dodecahedrons, the authors identified a series of partners, all of which have common structural patterns called “WW domains” [40]. These domains get their name from two conserved tryptophans (W) and are known to interact with PPxY patterns. Remarkably, two PPxY motifs (x meaning any amino acid) are conserved in the N-terminal of human penton bases, thus explaining why these partners interacted both with Pt-Dd and with Bs-Dd in the previous study. These cellular partners (WP1, WP2, and AIP4) are part of the ubiquitin ligase family, suggesting for the first time that these enzymes could play a role in the cycle of non-enveloped viruses. Previously, these enzymes were only reported in the budding of enveloped viruses [41]. Although dodecahedrons were used for the identification of these ubiquitin ligases, the dodecamerization is not necessary for the interaction since the HAdV2 penton base alone is also capable of recognizing these partners.

The last example to date is the use of dodecahedrons to identify the missing receptor for the subgroup B adenoviruses. Indeed, although it has been known for a long time that certain subgroup-B adenoviruses interact with CD46, it was also obvious that a certain number of them also interacted with an unidentified receptor X [17,42]. In 2011, Pt-Dd with 12 fibers (as in the native Ad3 virus) was used to fish for this missing receptor [18]. By affinity capture of HeLa cell proteins solubilized by Brij 96V detergent, it was possible to show that a high molecular weight band interacted specifically with the fiber. This interaction required the presence of calcium as already reported in 1995 [43]. The identification of this high molecular weight band by mass spectrometry demonstrated that desmoglein 2 (DSG2), a component of desmosomes, was the receptor for serotypes 3, 7, 11, and 14 [18]. Desmosomes are located at cellular junctions. However, the presence of this receptor in a region that is difficult for the virus to reach is far from an isolated case (for review, see Mateo et al. [44]). Interestingly, identification of desmoglein 2 highlighted a role played by Pt-Dd in the adenoviral replication cell cycle described below.

## 6. Biological Functions of the Dodecahedron ‘Pseudovirus’

While recombinant dodecahedrons can be produced with or without the fiber, only dodecahedrons displaying the fiber have been reported during the viral lifecycle [21]. As stated above, it has been reported that HAdV3 dodecahedrons are produced in large excess compared to progeny virions [30]. Therefore, the vast majority of penton bases going back to the nucleus after translation are used for dodecahedron formation, and only a small portion is used for neo-virion formation. Such an energy expenditure suggests that dodecahedrons must be involved to some extent in the viral replication strategy and could even be critical. Inside the cell, little is known except that these particles tend to accumulate on the inner side of the nuclear membrane away from the neo-synthesized virions [30]. During synthesis, they could also interact with several members of the ubiquitin ligase family, but this field remains to be studied further.

Outside the cell, the role of dodecahedrons may seem counter-intuitive at first glance. Indeed, Pt-Dd are composed of the two proteins involved in viral entry, and therefore represent a competitor for neo-synthesized virions. This competitor role could be demonstrated on A549 pneumocytes cells in culture infected with the HAdV3-GFP virus, where it was demonstrated that the dodecahedrons had an inhibitory effect on the infection in a dose-dependent manner [18]. However, in this study, the effect of the particles on cells was not limited to a simple competitive role. Rather, they also induced a significant transient cellular remodeling. This effect was measured by real-time cell impedance experiments. The magnitude of the impedance is dependent on the number of cells (cell index), as well as on the shape of the cells, and the cell-substrate attachment quality. When Pt-Dd were added to A549 cell cultures, a rapid and significant drop in impedance could be measured, thus demonstrating a rapid remodeling of the cell shape upon binding. In a similar experiment carried out with Bs-Dd, no effect was detected, thus shedding light on the role of the fiber in inducing this remodeling [45]. Moreover, when the experiment was done with the isolated Ad3 fiber (i.e., not on the dodecahedrons), no remodeling could be induced. This experiment shows, for the first time, the critical role of dodecamerization in inducing physiological effects—the constellated fibers on the dodecahedron produce an effect that cannot be mimicked without this scaffolding. This effect was also visualized by immunofluorescence studies on both fixed and live cells confirming that Pt-Dd induces a rapid (first effect visible in 10–20 min) and specific cell remodeling. Strikingly, both Bs-Dd and Pt-Dd were internalized in this experiment, suggesting that the event triggering cell remodeling is not induced by particle internalization but more likely earlier, when the fibers interact with DSG2. These results are in accordance with experiments made during the identification of DSG2 showing that Pt-Dd induces an epithelial to mesenchymal transition (EMT), resulting in the loss of epithelial markers, such as E-cadherin, and the gain of mesenchymal properties, like cytoskeleton rearrangement, migratory potential, and matrix metalloproteases (MMPs) expression [18,46]. The mechanism underlying this process was investigated later on. A MAP kinase activation triggered by either Pt-Dd or HAdV3 binding to DSG2, results in the activation of the ADAM17 matrix-metalloproteinase and in the shedding of DSG2 (Figure 5a) [47]. In the scenario where cell disruption by excess capsomer has already been reported for HAdV2 upon ‘isolated’ fiber binding to the CAR receptor (a member of “tight junctions” between the cells) [48], the HAdV3 fiber must be dodecamerized to exert such an effect. This was demonstrated with an HAdV3 mutant (mu-Ad3GFP) where the penton bases were modified to suppress decodecamerization (D100 and R425 mutations). The latter mutation was unable to trigger cell remodeling and featured a worse spreading towards the center of cellular spheroids [49].

Another role of the dodecahedrons upon release from the infected cell, is to trap defensins and, more specifically, the Human Defensin 5 (HD5). HD5 is a small antimicrobial peptide which is able to neutralize the HAdVs from the A, B, C, and E species. To do so, HD5 binds to the HAdV’s capsids, creating a bridge that stabilizes the penton base and the fiber, therefore blocking the pVI protein exposition during the virus uncoating [50,51]. This impairs pVI presentation to the membrane of the endosome, which traps the virus after internalization [52]. The virus is then unable to replicate and is therefore neutralized. The involvement of dodecahedrons in this mechanism has also been studied. It has been shown that HAdV3 dodecahedrons are able to trap and neutralize HD5 secreted by cancer cells, thus acting as a decoy (Figure 5b) [53]. This allows HAdV3 to spread, whereas, in the same conditions the non-dodecahedron forming HAdV5 spreading was limited. Like HAdV5, ‘mu-Ad3GFP’ expressing non-dodecamerizing HAdV3 penton bases was also unable to trap defensins, showing that dodecamerization is critical in this process.

## 7. Biotechnological Applications

### 7.1. DNA Delivery

As stated above, dodecahedrons can be rapidly internalized in cells and do not contain any pre-existing nucleic acids. Therefore, they can be excellent vectors for delivering a gene of interest into cells. However, dodecahedrons do not possess an internal cavity large enough to incorporate nucleic acids. Indeed, this central cavity of around 350 nm^3^ cannot theoretically enable the packing of more than 100 nucleotides. To overcome this issue, a bifunctional peptide has been designed [31]. This peptide mimics the first 20 amino acids of the trimeric HAdV3 fiber and has a polylysine tail of 20 residues. This peptide can interact with the penton base via the fiber amino acids, as well as with DNA via an electrostatic interaction between the lysine NH_3_^+^ groups and the DNA PO_3_^−^ groups. 

Pt-Dd and Bs-Dd were incubated with the bifunctional peptide and a DNA plasmid carrying the luciferase gene (Figure 6a). It was demonstrated that these complexes could efficiently enter cells and transfer the DNA plasmid into the cell where it reached the nucleus and resulted in gene expression. It is worth noticing that the level of gene expression (and thus of gene transfer) was higher with the penton base dodecahedron. This is probably due to the fact that 3 out of the 5 fiber binding sites on each pentameric penton base are blocked when the trimeric fiber is present, thus limiting access to the fiber-mimicking peptide. However, unlike the use of a ‘classical’ viral vectors, in this kind of approach, the genetic material is not protected. In a similar work using the non-dodecamerizing HAdV5 penton base fused to a decalysine (called 3PO), it has been clearly shown that addition of protamine protected DNA from serum nucleases [54]. Such system has been further explored by adding a heregulin ligand to bind with high affinity Her2/3 or Her2/4 overexpressed on certain aggressive breast cancers [55] and a recent work reported the delivery of RNAi in HER3+ tumors in vivo [56].

### 7.2. Protein Delivery

For the same reasons enounced previously, dodecahedrons represent serious candidates for protein delivery. A study conducted in 2003 demonstrated that dodecahedrons are able to vectorize proteins inside cells [32]. HAdV3 Bs-Dd and Pt-Dd decorated with non-neutralizing monoclonal antibodies (MAbs) directed against the penton base were incubated with cells. In only a few min, antibodies could be delivered inside the cells with both Bs-Dd and Pt-Dd, although the latter was more efficient. Since the MAb alone was not able to be internalized by cells, this data highlights the capacity of dodecahedrons to allow transduction of a high molecular weight protein into cells. However, this promising technique only applies to monoclonal antibodies directed against a component of the dodecahedron. In order to transport any protein of interest into the cells, a more versatile system was necessary.

Therefore, a more universal system has been developed using the interaction properties of the highly-conserved motif “PPxY” motif of the penton base and its “WW” domains (described in Section 5). Three out of the four “WW” domains of the ubiquitin ligase Nedd4 were expressed in fusion with the reporter protein Maltose Binding Protein (MBP), and this protein was efficiently delivered to cells via dodecahedrons (Figure 6a) [57]. This system paved the way to the development of new vaccine technologies described below (cf. ‘vaccination’).

## 8. Therapeutic “Junction” Opener Effect Improving Anti-Tumor Drugs Efficiency 

When the DSG2 was first identified in 2011, it was observed that Pt-Dd binding to DSG2 triggered an epithelial-to-mesenchymal transition (EMT) on breast cancer cells (B474), leading to transient opening of intercellular junctions [18]. This transient opening allowed exposition of poorly accessible receptors, which could be of therapeutic importance but are partially hidden between two adjacent cells. This is the case for Her2/neu, which is targeted by the widely used monoclonal antibody (Mab) ‘Herceptin’ in breast cancer treatment (Figure 6b). It has been shown that the EMT triggered by recombinant Pt-Dd allowed a better exposition of the Her2/neu receptor in tumors. Indeed, a stronger and faster decrease of the tumor size was observed in the group of mice treated with Herceptin combined with Pt-Dd, compared with the group injected with the therapeutic MAb alone. This study demonstrates that dodecahedrons could be used as therapeutic adjuvants in order to improve the efficiency of anti-tumor treatments. This work served as the basis for the development of a new set of molecules with multimerized HAdV3 fibers called ‘junction openers’ (JO). These have been tested in preclinical studies with other therapeutic MAbs, like the anti-EGF ‘cetuximab’ and chemotherapeutic agents, such as the liposomal doxorubicin ‘Doxil’ [58,59,60].

### Vaccine Development 

Most of the preventive vaccines against viral diseases currently in use against are inactivated (chemically or thermally) or attenuated (natural or artificial virulence loss) live viruses. As they are real viruses, there is always a possibility of reactivation. This is not the case for dodecahedrons, which do not contain any viral genetic material. Additionally, the multivalency of dodecahedrons makes them a good tool to trigger an efficient immune response as previously reported for other virus-like particles (VLPs) [61,62].

Several studies have been conducted to evaluate the potential of dodecahedrons as vaccines. In a first approach, 3 out of the 4 Nedd4 “WW” domains were fused to the ovalbumin model antigen _248_OVA_376_ cargo. This fusion protein was then attached to the dodecahedrons through the WW-PPxY interaction, so that the cargo was presented in high numbers of copies (Figure 6c) [63,64]. Naive mice were then immunized with the WW-OVA dodecahedrons, which lead to a specific CD8+ T lymphocyte response against OVA. These CD8+ T lymphocytes showed a specific cytotoxic effect against melanoma cells expressing the OVA antigen on their surface (B16-OVA). These promising results paved the way to more experiments with ovalbumin antigens. Mice were immunized with 2 injections of WW-OVA dodecahedrons 14 and 7 days before injection of melanoma cells B16 expressing the OVA antigen (B16-OVA). The tumor growth was evaluated and only one mouse out of 10 developed a tumor. However, it is worth noticing that, for this mouse, the tumor development was delayed (D25 post injection) compared to B16-OVA non-immunized group (around 10 days post injection). The other 9 mice survived several months after the tumor injection, whereas, in the negative control group, the overall survival did not exceed 20 days.

These first results with the model antigen ovalbumin highlighted the strong potential of using dodecahedrons in vivo in the vaccine field and in cancer immunotherapy. A similar in vitro study was later performed in infectiology with the influenza hemagglutinin attached, showing that several applications could be addressed by using this strategy [65]. However, this kind of approach suffered from two major shortcomings: (i) the presence of an adapter of human origin (the WW domain) and (ii) the need for two different expression systems. To overcome these issues, a new system has recently been reported. The HAdV3 penton base gene was engineered by insertion of restriction sites to create three independent sites for epitope display: one in the VL loop and two others in the RGD loop (flanking the preserved RGD motif) of the penton base sequence. Therefore, various antigenic epitopes of choice could be inserted into this plug-and-play multiepitope display platform named ADDomer (Figure 6c). To illustrate the potential of such a system, the chikungunya major neutralizing epitope E2EP3 (Epitope 3 of the chickungunya E2 protein, 18 amino acids), which is linearly displayed at the new N terminus of E2 glycoprotein created after furin cleavage, was inserted into the ADDomer. This created a virus-like-particle (VLP) vaccine candidate named ADDomer-tevCHIK. To loosen this conformation, a highly specific protease site (Tobacco Etch virus: TEV) was inserted upstream of the epitope. Upon cleavage, the epitope can readopt its native-like conformation comprising an exposed N-terminal serine. After injection in mice, no E2EP3-specific antibodies were obtained with the uncleaved ADDomer-tevCHIK. On the contrary, cleaved ADDomer-tevCHIK exposing the native-like E2EP3 led to a very strong humoral response in absence of adjuvant [66]. In comparison, a nanoparticle scaffold made of polylactic acids (PLA) displaying the same amount of the E2EP3 epitope did not lead to an anti-E2EP3 immune response. This suggests that the ADDomer itself has an adjuvant effect. It is worth noticing that ADDomers rapidly drain to lymph nodes upon injection in mice and are efficiently internalized in immune blood cells, which is essential for a vaccine candidate. In this study, it was also shown that the ADDomer is remarkably thermostable, which represents a crucial asset for the deployment of ADDomer-based vaccines in remote countries. In summary, all of these results involving a real antigen from a pathogen confirmed the strong potential of dodecahedrons in the vaccine field.

## 9. Concluding Remarks

In this review, multiple facets of the adenovirus dodecahedron were detailed. It is interesting to note that these particles, identified in the late 60s by Erling Norrby, were not studied again until their recombinant expression 30 years later. One unique property of the dodecahedrons relies on the ability of the penton bases to interact together symmetrically while they are not in contact in the virion. The structural mechanisms underlying the formation of dodecahedrons have been elucidated over time, but certain aspects remain to be determined formally, such as where their assembly takes place in the cell and whether chaperone proteins help their assembly. Their production in large numbers suggests that these particles play a role in the viral lifecycle, and it has been confirmed that they favor spreading of the virus after lysis of the infected cells. However, it would be necessary to further explore their putative intracellular role in the infected cell before their dispersion in the extracellular medium. The strong internalization properties of these particles, which interact with the same receptors as the virus but also with other molecules, such as HSPGs, have been used for biotechnological applications. Their use could be further improved by increasing their endosomolysis capacity. Their applications in vaccinology seem promising, as these particles are particularly stable and the 60-mer controlled multimerization, allowing good recognition by the immune system. From a purely geometric point of view, it is interesting to note that an icosahedral virus is capable of producing its dual dodecahedral form. These forms, which intrigued Leonardo da Vinci because of their aesthetics and underlying mathematical relationships since the golden ratio intervenes in the calculation of their radius, their volume, and their coordinates, probably still have a lot to teach us.

## Figures and Tables

**Figure 1 viruses-12-00718-f001:**
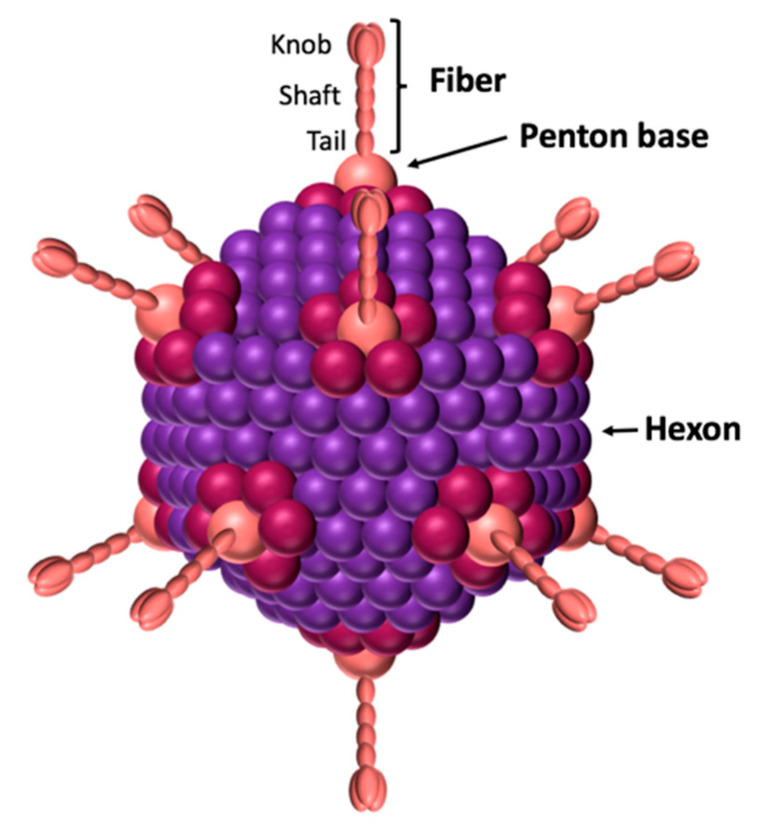
Schematic view of adenovirus. The icosahedral capsid is formed by the hexon. The penton base is located at the 12 vertices and forms a non-covalent complex with the trimeric fiber. The fiber’s knob domain is responsible for the interaction with the receptors.

**Figure 2 viruses-12-00718-f002:**
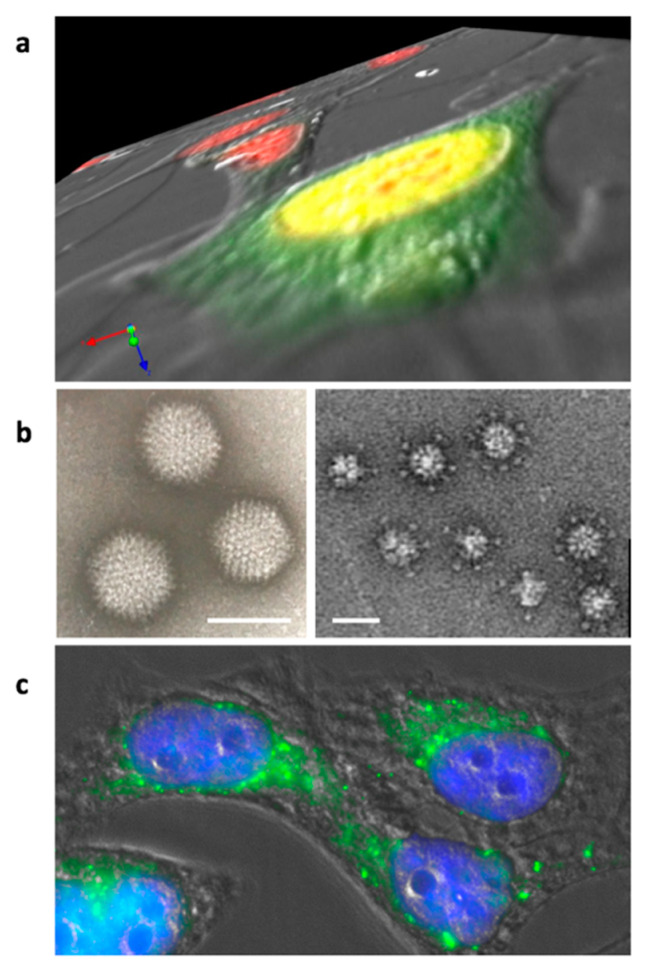
Adenovirus dodecahedron formation and internalization. (**a**) Z-series of Hela cells infected for 16 h by wt-HAdV3. Nuclei are stained in red and the penton base is detected in green. The penton base is synthesized in the cytoplasm and transported to the nucleus (yellow results from superposition of the red and green signals) where dodecahedron assembly takes place. (**b**) Electron microscopy images of the purified adenovirus and (Pt-Dd) ‘penton dodecahedrons’ (bars: 90 and 30 nm, respectively). (**c**) Hela cells incubated with recombinant Pt-Dd for 1 h. Cell shapes were observed by DIC, nuclei are stained in blue, and Pt-Dd are detected in green.

**Figure 3 viruses-12-00718-f003:**
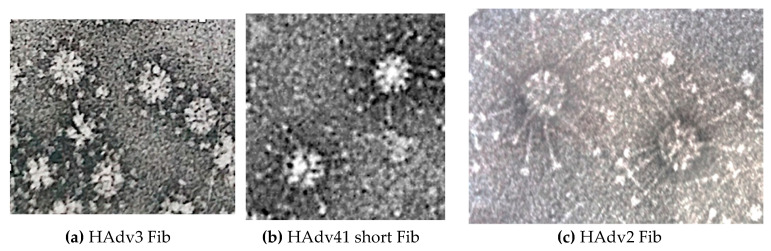
Recombinant HAdV3 dodecahedrons pseudotyped with different fibers. (**a**) Base Dodecahedron (Bs-Dd) coexpressed with its corresponding HAdV3 fibers, (**b**) the enteric HAdV41 short fiber, or (**c**) co-incubated with the HAdV2 fiber are observed by negative staining at the same scale (Bars: 30 nm).

**Figure 4 viruses-12-00718-f004:**
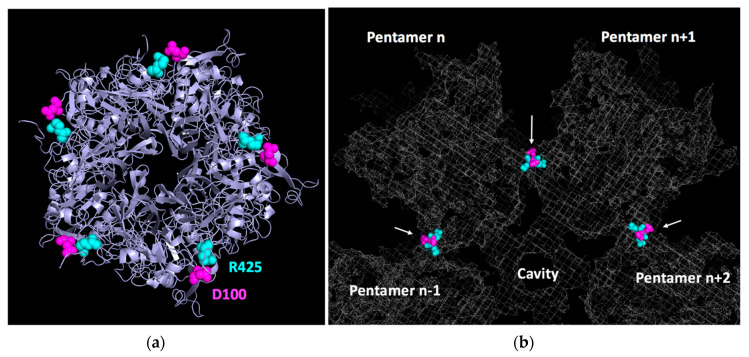
Contact between adjacent pentamers of a dodecahedron. (**a**) Bottom view of a single pentamer with D100 highlighted in magenta and R425 in light blue. (**b**) Zoomed-in slab view of a dodecahedron (grey mesh) showing contacts made by D100 and R425 from adjacent pentamers (PDB 6HCR.).

**Figure 5 viruses-12-00718-f005:**
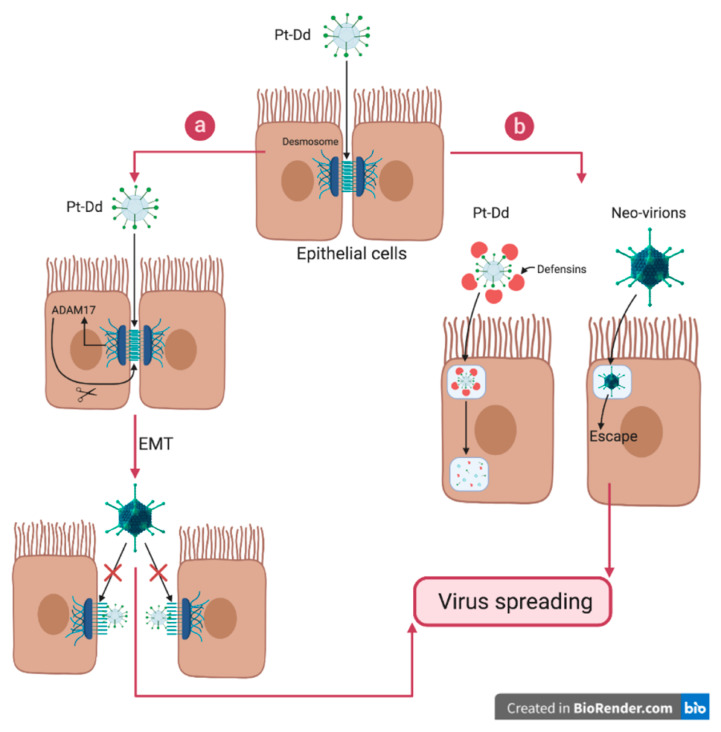
Biological function of the dodecahedron ‘pseudovirus’. Pt-Dd facilitates virus spreading by both interacting with DSG2 and acting as a decoy. (**a**) Pt-Dd binds to the virus receptor desmoglein 2 located in desmosomes and induces an epithelial to mesenchymal transition (EMT) through MAP kinase signaling and activation of the ADAM17 matrix-metalloproteinase. Moreover, competition between Pt-Dd and virions for DSG2 forces virions to spread in the tissue. (**b**) At the same time, Pt-Dd traps Human Defensin (HD)5, which is known to block adenovirus infection by stabilizing the capsid, thus preventing pVI release and subsequent endosomolysis. By acting as a decoy, Pt-Dd protects neo-virions from defensin attack. Figure created with BioRender.com.

**Figure 6 viruses-12-00718-f006:**
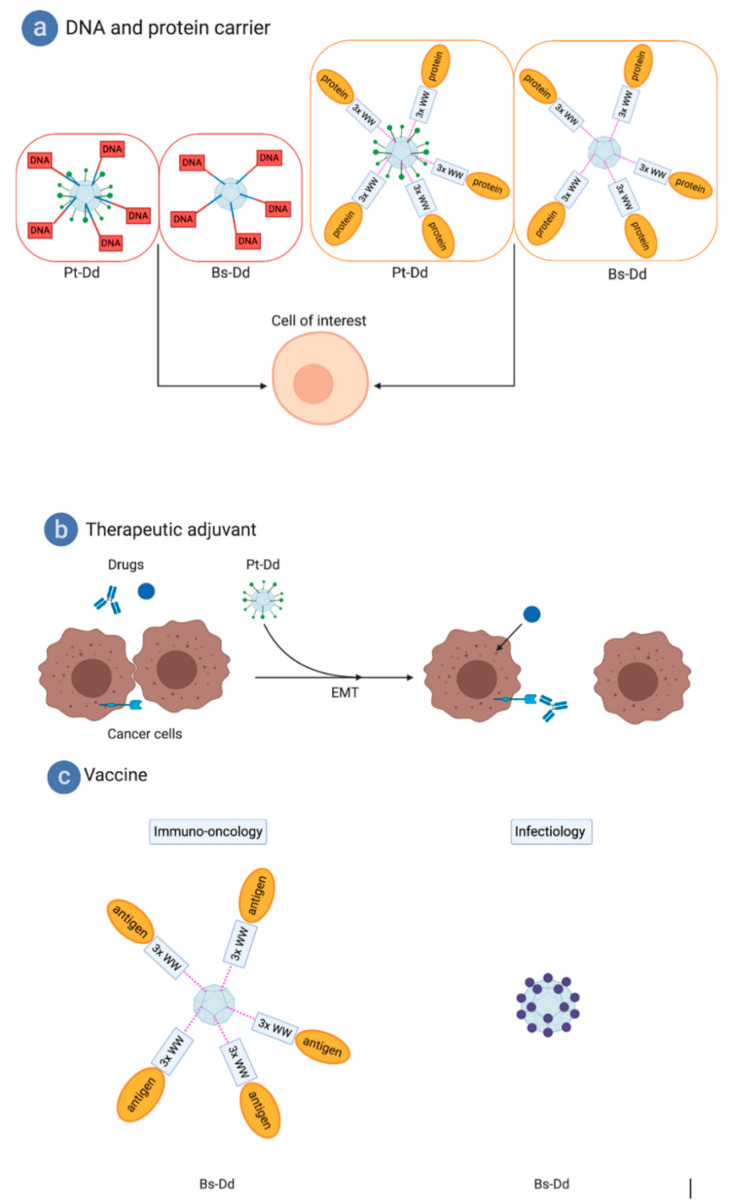
Biotechnological applications of the dodecahedron ‘pseudovirus’. (**a**) The dodecahedron ‘pseudovirus’ can be used to deliver DNA due to a bispecific peptide or proteins of interest fused to the WW domains. (**b**) The dodecahedron ‘pseudovirus’ can be used as a therapeutic adjuvant by triggering cell remodeling by making hidden targets accessible to therapeutic monoclonal antibodies (MAbs) or by enabling better access to chemotherapeutic compounds. (**c**) For vaccine purposes, cancer antigens fused to WW domains can be grafted to Bs-Dd. Alternatively, a single particle can be produced to display epitopes from pathogens, such as emergent viruses. Figure created with BioRender.com.

**Table 1 viruses-12-00718-t001:** The five platonic solids. Their shapes and features are reported, as well as their duality and symbolic meanings.

Name	Cube	Octahedron	Tetrahedron	Icosahedron	Dodecahedron
**Shape**	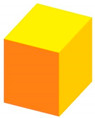	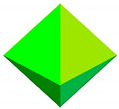	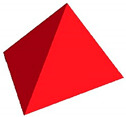	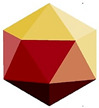	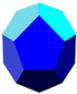
**Features**	6 faces8 vertices12 edges	8 faces6 vertices12 edges	4 faces4 vertices6 edges	20 faces12 vertices30 edges	12 faces20 vertices30 edges
**Facets**	Squares	Equilateral triangles	Equilateral triangles	Equilateral triangles	Pentagons
**Duality**	Dual 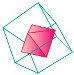 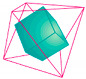	Self-dual 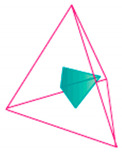	Dual 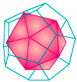 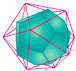
**Symbol**	Earth	Air	Fire	Water	Universe/Heaven

**Table 2 viruses-12-00718-t002:** Subgroups and dodecahedron formation. human adenovirus (HAdV) serotypes belonging to different subgroups are classified according to dodecahedron formation. Note that dodecahedrons have never been observed in the well-studied subgroup C and that subgroup B contains both dodecahedron-forming and non-dodecahedron forming serotypes.

Subgroup	A	B	C	D	E	F	G
Dd forming		3, 7, 11		9, 15	4		
Non Dd forming	12	16	1, 2, 5, 6			40, 41

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
