# Peer review of "The Adenovirus Dodecahedron: Beyond the Platonic Story"

_viruses, 2020, doi:10.3390/v12070718_

Round 1

Reviewer 1 Report

This review article provides a comprehensive overview of the research findings and applications of penton base dodecahedrons (Dd), including a brief history of their discovery and very interesting discussion of the naturally occurring geometry giving rise to these structures. The authors themselves have been the major discoverers and developers of the Dd technology over the years and thus self-citation in this context is not necessarily inappropriate; although it would have been objectively more complete if similar technologies (especially those with findings that further support the functional activities of the penton base) could be acknowledged. The slightly philosophical discussion on the geometries occurring through the fit and assembly of basic units was an intriguing departure that offers an appreciation of the simple mathematical elegance underlying these structures; this discussion seems completely appropriate in this review, and an enjoyable read. The figures are beautiful and enhance the article.

Some notes to consider:

  1. Lines 77-79, referrring to Plato's observations: it would be helpful to include citations.
  2. Line 170-171, referring to VLPs: again, citations could be helpful here.
  3. Lines 330-332, referring to gene transfer: It would be useful to include a few sentences briefly acknowledging additional limitations of this approach for translational application with appropriate citations; i.e. shows ability to deliver nucleic acids in principle, but would require additional  reagents to protect/condense cargo for in vivo application (with appropriate citations); also gene transfer/expression ability supports the contention that Dd (and penton base for that matter) facilitate transit across cellular and intracellular barriers (with appropriate citations).

Author Response

Answer to Reviewer 1:

This review article provides a comprehensive overview of the research findings and applications of penton base dodecahedrons (Dd), including a brief history of their discovery and very interesting discussion of the naturally occurring geometry giving rise to these structures. The authors themselves have been the major discoverers and developers of the Dd technology over the years and thus self-citation in this context is not necessarily inappropriate; although it would have been objectively more complete if similar technologies (especially those with findings that further support the functional activities of the penton base) could be acknowledged. The slightly philosophical discussion on the geometries occurring through the fit and assembly of basic units was an intriguing departure that offers an appreciation of the simple mathematical elegance underlying these structures; this discussion seems completely appropriate in this review, and an enjoyable read. The figures are beautiful and enhance the article.

We warmly thank the reviewer for these very positive comments. We have now added a paragraph dealing with a similar technology based on the Ad5 penton, which reinforces the interest and completeness of this review (lines 339-346).

Some notes to consider:

  1. Lines 77-79, referrring to Plato's observations: it would be helpful to include citations.

An internet link was added (line 85)

  1. Line 170-171, referring to VLPs: again, citations could be helpful here.

A comprehensive review on VLP is now cited (Zeltin 2013, line 175)

  1. Lines 330-332, referring to gene transfer: It would be useful to include a few sentences briefly acknowledging additional limitations of this approach for translational application with appropriate citations; i.e. shows ability to deliver nucleic acids in principle, but would require additional  reagents to protect/condense cargo for in vivo application (with appropriate citations); also gene transfer/expression ability supports the contention that Dd (and penton base for that matter) facilitate transit across cellular and intracellular barriers (with appropriate citations).

As stated above, we found that it was a good idea to integrate the very nice works on the HAdV5 penton bases performed by Pr Medina-Kauwe’s Lab, reporting the different limitations of this kind of technology and how to deal with.

We are grateful for your positive reviewing and your constructive remarks.

Reviewer 2 Report

General Comments

The review describes the dodecahedron structure discovered for adenoviruses and their application potential in biotechnology and vaccine development. The manuscript is generally well written and provides an interesting view of the structural aspects of virions and how the dodecahedron can be employed as delivery vehicles. However, before being accepted for publication, I suggest a minor revision of the manuscript taking into account the comments/suggestions presented below.

There are hardly any references in the first paragraph of the Introduction.

For the reader not directly working with adenoviruses it would be informative to include a figure of the schematic structure of adenoviruses.

It is recommended to use “viral” instead of “virus’s” (also “adenovirus’s”) throughout the manuscript.

In section 8.1 on vaccine, the authors claim that inactivated or attenuated live viruses are most commonly used for vaccine development. This is the case for vaccines against pathogenic viruses, but not cancer vaccines, which is what the authors next give an example of.

Specific Comments

L36: “responsible of” > “responsible for”

L42: “All the adenoviruses” > “All adenoviruses”

L59: “to the other” > “to another”; “9nm” > “9 nm”; “36nm” > “36 nm”

L81: “of the other” > “of another”

L99: “nM” refers to nanomolar, should be “nm”

L109-113: The text is in the middle of Table II

L115: “immunofluorescent detection” > “immunofluorescence”

L117: “20h” > “20 h”

L126: “Chapter” > “Section”

L136: “This” > “The”

L143: “enter the cell” > “enter cells”

L145: “into the cells” > “into cells”

L149: “16H” > “16 h”

L158: “[31]or” > “[31] or”

L160: “30nm” > “30 nm”

L169: “wen” > “when”

L191: “Later-on” > “Later on”

L215: Add comma after “dodecamers”

L234: “aminoacid” > “amino acid”

L244: There is no reference for “In 2011….”

L245: “cells proteins” > “cell proteins”

L250: “14[18]” > “14 [18]”

L256: “viral cycle” > “viral lifecycle”

L257: Please define “neo”

L324: “350nm3” > “350 nm3

L343: What dose “large protein transduction” refer to? Is it a question of transduction of large proteins or large amounts of proteins?

L345: I assume the authors mean “transport into the cells”

L350: “delivered within cells” > “delivered to cells”

L353: “Therapeutic_”junction” opener” > “Therapeutic “junction” opener”

L368: “In vaccination” > “Vaccine development”

L370: “virus” > “viruses”

L377: “high copies” > “high numbers of copies”

L425: “these particle” > “these particles”

L431: “viral cycle” > “viral lifecycle”

Author Response

Answer to Reviewer 2:

The review describes the dodecahedron structure discovered for adenoviruses and their application potential in biotechnology and vaccine development. The manuscript is generally well written and provides an interesting view of the structural aspects of virions and how the dodecahedron can be employed as delivery vehicles. However, before being accepted for publication, I suggest a minor revision of the manuscript taking into account the comments/suggestions presented below.

We warmly thank the reviewer for these very positive comments, your suggestions to improve this review and the list of corrections that was really helpful to us.

There are hardly any references in the first paragraph of the Introduction.

We agree and we added a link to https://sites.google.com/site/adenoseq/ which is very informative to people wanting to know more on adenovirus classification.

For the reader not directly working with adenoviruses it would be informative to include a figure of the schematic structure of adenoviruses.

A new figure (Figure 1) is now added.

It is recommended to use “viral” instead of “virus’s” (also “adenovirus’s”) throughout the manuscript.

Done!

In section 8.1 on vaccine, the authors claim that inactivated or attenuated live viruses are most commonly used for vaccine development. This is the case for vaccines against pathogenic viruses, but not cancer vaccines, which is what the authors next give an example of.

That is true, we now specify ‘preventive vaccines against viral diseases’ (line 380)

Specific Comments:

All are done (in yellow in the manuscript). We highligted in bald our answers to specific points below.

L36: “responsible of” > “responsible for”

L42: “All the adenoviruses” > “All adenoviruses”

L59: “to the other” > “to another”; “9nm” > “9 nm”; “36nm” > “36 nm”

L81: “of the other” > “of another”

L99: “nM” refers to nanomolar, should be “nm”

L109-113: The text is in the middle of Table II (It has been been processed by the editorial office not by us, we will check the galley proofs to ensure that the problem is solved before publication)

L115: “immunofluorescent detection” > “immunofluorescence”

L117: “20h” > “20 h”

L126: “Chapter” > “Section”

L136: “This” > “The”

L143: “enter the cell” > “enter cells”

L145: “into the cells” > “into cells”

L149: “16H” > “16 h”

L158: “[31]or” > “[31] or”

L160: “30nm” > “30 nm”

L169: “wen” > “when”

L191: “Later-on” > “Later on”

L215: Add comma after “dodecamers”

L234: “aminoacid” > “amino acid”

L244: There is no reference for “In 2011….”

L245: “cells proteins” > “cell proteins”

L250: “14[18]” > “14 [18]”

L256: “viral cycle” > “viral lifecycle”

L257: Please define “neo” We changed by progeny (line 261)

L324: “350nm3” > “350 nm3

L343: What dose “large protein transduction” refer to? Is it a question of transduction of large proteins or large amounts of proteins? We changed by ‘a high molecular weight protein’ (line 354)

L345: I assume the authors mean “transport into the cells”

L350: “delivered within cells” > “delivered to cells”

L353: “Therapeutic_”junction” opener” > “Therapeutic “junction” opener”

L368: “In vaccination” > “Vaccine development”

L370: “virus” > “viruses”

L377: “high copies” > “high numbers of copies”

L425: “these particle” > “these particles”

L431: “viral cycle” > “viral lifecycle”

We are grateful for your positive reviewing, your suggestions and the list of corrections.